



# Multi-decadal analysis of root-zone soil moisture applying the exponential filter across CONUS

Kenneth J. Tobin[1], Roberto Torres[1], Wade T. Crow[2], and Marvin E. Bennett[1]

[1] Texas A&M International University, Center for Earth and Environmental Studies, Laredo, Texas, United States

[2] United States Department of Agriculture, Agricultural Research Service Hydrology and Remote Sensing Laboratory, Beltsville, Maryland, United States

*Correspondence to*: Kenneth J. Tobin (ktobin@tamiu.edu)

**Abstract.** This study applied the exponential filter to produce an estimate of root-zone soil moisture (RZSM). Four types of microwave-based, surface satellite soil moisture were used. The core remotely sensed data for this study came from NASA's long lasting AMSR-E mission. Additionally three other products were obtained from the European Space Agency Climate Change Initiative (CCI). These datasets were blended based on all available satellite observations (CCI-Active; CCI-Passive; CCI-Combined). All of these products were quarter degree and daily. We applied the filter to produce a soil moisture index (SWI) that others have successfully used to estimate RZSM. The only unknown in this approach was the characteristic time of soil moisture variation (T). We examined five different eras (1997-2002; 2002-2005; 2005-2008; 2008-2011; 2011-2014) that represented periods with different satellite data sensors. SWI values were compared with *in situ* soil moisture data from the International Soil Moisture Network at a depth ranging from 20 to 25 cm. Selected networks included the U.S. Department of Energy Atmospheric Radiation Measurement (ARM) program (25 cm), Soil Climate Analysis Network (SCAN; 20.32 cm), SNOwpack TELemetry (SNOTEL; 20.32 cm), and the U.S. Climate Reference Network (USCRN; 20 cm). We selected *in situ* stations that had reasonable completeness. These datasets were used to filter out periods with freezing temperatures and rainfall using data from the Parameter elevation Regression on Independent Slopes Model (PRISM). Additionally, we only examined sites where surface and root zone soil moisture had a reasonable high lagged correlation coefficient (r>0.5).

The unknown T value was constrained based on two approaches: optimization of root mean square error (RSME) and calculation based on the NDVI value. Both approaches yielded comparable results; although, as to be expected, the optimization approach generally outperformed NDVI based estimates. Best results were noted at stations that had an absolute bias within 10%. SWI estimates were more impacted by the *in situ* network than the



surface satellite product used to drive the exponential filter. Average Nash-Sutcliffe coefficients (NS) for ARM
ranged from -0.1 to 0.3 and were similar to the results obtained from the USCRN network (0.2 to 0.3). NS values
from the SCAN and SNOTEL networks were slightly higher (0.1 to 0.5). These results indicated that this approach
had some skill in providing an estimate of RZSM. In terms of root mean square error (RMSE; in volumetric soil
moisture) ARM values actually outperformed those from other networks (0.02 to 0.04). SCAN and USCRN RMSE
average values ranged from 0.04 to 0.06 and SNOTEL average RMSE values were higher ranging (0.05 to 0.07).
These values were close to 0.04, which is the baseline value for accuracy designated for many satellite soil moisture
missions.
**1 Introduction**
Soil moisture is one of the most difficult hydrologic variables to either monitor or model (Lattenmaier et al., 2015).
Understanding soil moisture dynamics is critical to support many diverse applications in hydrology, meteorology,
and agriculture. In the agricultural sector a fundamental limiting factor that constrains crop productivity is root zone
soil moisture (RZSM). Understanding root zone moisture dynamics is important also from a water resource
standpoint and is a valuable measure in drought monitoring (Bolten et al., 2010; Bolten and Crow, 2012). The
dimensions of RZSM also impact other systems beyond the hydrologic cycle, most notably with the quantification
of carbon fluxes within soils. Therefore, direct sensing of RZSM dynamics will bring us closer to a truer
understanding of the carbon soil pool, with obvious implications for future climate change.

Given the importance of RZSM to agricultural and other applications, more effort is needed to understand the

impacts of climate change associated with this critical variable. The National Aeronautics and Space Administration
(NASA), European Space Agency (ESA), and other governments across the world have had a long history of
supporting missions that generate remotely sensed surface soil moisture, including the Scanning Multichannel
Microwave Radiometer (SMMR), the Special Sensor Microwave Imager (SSM/I), Tropical Rainfall Measurement
Mission (TRMM), Advanced Microwave Scanning Radiometer-Earth Observing System (AMSR-E), Soil Moisture
and Ocean Salinity (SMOS), Soil Moisture Active Passive (SMAP), scatterometers on the European Remote
Sensing satellites, which includes (SCAT) and the Advanced Scatterometer (ASCAT) to name only a few (*e.g.*
Lakshmi et al. 1997; Wagner et al. 1999; Kerr et al. 2001; Jackson et al. 2002; Hutichson, 2003; Njoku et al, 2003;
McCabe et al. 2005; Owe et al., 2008; Entekhabi et al., 2010). Passive microwave soil moisture estimate, like



AMSR-E measured the horizontal and vertical polarization temperatures in several wavelengths, which include:
6.6/6.9 GHz (C-band), and 10.7 GHz (X-band), 19.3 GHz (Ku-band). In addition, the vertical polarization is
examined at 36.5/37.0 GHz (Ka-band). An advantage of the more recent SMOS and SMAP missions is that they
operate at a lower frequency 1.2/1.4 GHz (L-band), which has great penetrative power, especially in highly
vegetated areas. In terms of the active sensors both SCAT and ASCAT operated at 5.3 GHz (C-band) and have a
similar design philosophy. These sensors make sequential observations of the backscattering coefficient with six
sideways looking antennas and make sequential observations of the backscattering coefficient using three polarizing
antennas.
Liu et al, (2012) described the development of two extensively validated surface soil moisture products.
These products were created using a harmonized dataset based on all available soil moisture retrievals; one from the
Vienna University of Technology (TU Wien) based on active microwave observations (Wagner *et al.,* 2003, Bartalis
*et al.,* 2007) and one from the Vrije Universiteit Amsterdam (VUA), in collaboration with NASA Goddard Space
Flight Center Hydrological Sciences Laboratory, based on passive microwave observations (Owe *et al.,* 2008). This
effort was a part of the ESA Climate Change Initiative (CCI). The harmonization of these datasets incorporated the
advantages of both microwave techniques and spanned the entire period from 1978 onward. This effort is unlike
NOAA's Soil Moisture Operational Products System (SMOPS), which was a long-term record of soil moisture
based on only passive microwave data.
A long-standing goal of the soil remote sensing community is to develop techniques that can observe changes
in RZSM at depths greater than 10 cm, because all of the missions described above are confined to sensing moisture
only within the top 5 cm of the profile. In 2015 NASA launched the SMAP mission that had the potential to
combine of the advantages of passive and active microwave retrievals to estimate soil moisture dynamics at depth.
Unfortunately, early during this mission the satellite's radar failed. Despite this setback NASA had invested
considerable resources into the development of an Ensemble Kalman Filter (EnKF)-based Level 4 RZSM product
for SMAP (Reichle et al., 2016) and the development of lower-frequency airborne radar systems for deeper
penetration of the soil column (via the EV-1 AirMOSS project). While this work is to be commended, the limited
time availability of these products precludes their use for long-term climatic trend studies.
This study used the exponential filter to leverage the longer duration CCI surface soil moisture record to
produce a record of RZSM that can be compared over almost two decades (1997-2014). Wagner et al. (1999)



developed the exponential filter to examine soil moisture trends from ERS Scatterometer data focusing on the
Ukraine. A later refinement of this filter included the development of a recursive version that had the virtue of a
greater ease of implementation (Albergel et al, 2008). In recent years several authors have produced RZSM
estimates using the exponential filter and have conduct comparisons at a range of spatial scales (Ford et al. 2014;
Manfredaet al. 2014; Qiu et al. 2014; Peterson et al. 2014; Kedzior and Zawadzki, 2016). At the heart of the
exponential filter method is the assumption of hydrologic equilibrium within the soil profile that makes it possible to
estimate RZSM by using only surface measurements, provided that soil physical properties are known. This method
also assumes that there is no loss from the root zone due to transpiration. Transfer of soil moisture from the surface
to the root zone is controlled by a pseudodiffusivity term that allows both positive and negative fluxes from and to
the deep layer. This approach overcame a limitation of the EnKF approach in that data assimilation is not dependent
on obtaining data from a land surface model, in which there can be significant uncertainty in terms of the model
parameters used to constrain water and energy balances (Kumar et al, 2009). This study presents the results of the
application of the exponential filter produced using four satellite soil moisture products from 1997-2014 focusing on
Continental United States (CONUS). As such this work represents a unique application of the exponential filter over
a mutlidecadal time scale, which is only afforded by the long duration CCI record.
**2 Data**
**2.1 Era Definitions**
The data examined in this study spans over 17 years. As such we compared soil moisture produced by the
exponential filter over five, roughly equal eras (3-4.5 year), which were defined based on the available satellite
retrievals during each era (see Liu et al. 2012). These eras included: November 27 1997-June 18 2002 (pre-AMSR-
E), June 19 2002-June 30 2005 (Early AMSR-E), July 1 2005-June 30 2008 (Middle AMSR-E), July 1 2008-
October 3 2011 (Late AMSR-E), and October 4 2011-December 31, 2014 (post-AMSR-E). The pre-AMSR-E era
relied heavily on the TRMM Microwave Imager (TMI) passive observations and SCAT active retrievals that
operated until 2006. In fact, the climatology of the passive dataset during this period was rescaled based on TMI
data and likewise the same was true of AMSR-E during eras 2-4. During the Early AMSR-E era passive
observations from the Windsat satellite came on line (Gaiser 2004). The Middle AMSR-E era was a time of
transition in terms of active observations as the SCAT satellite is replaced by ASCAT. The Late AMSR-E era saw



the arrival of the ESA SMOS mission. After the failure of AMSR-E, SMOS observations took on a more prominent
role within the CCI passive microwave framework. Also during the post-AMSR-E the Japanese Space Agency
launched AMSR2 (Wentz et al. 2014), which is considered the replacement for the long lasting AMSR-E mission.
**2.2 In Situ Soil Moisture**
Direct, *in situ* comparisons were made between RZSM estimates with *in situ* data from the International Soil
Moisture Network (ISMN; Dorigo et al., 2011). The ISMN provides access to a host of meteorological and soil
moisture data (at many depths). In this study, we selected soil moisture at two depths. Surface soil (0-10 cm) and
RZSM (20-25 cm) moisture was compared to assess the performance of the exponential filter method. In this study
we focused on four networks within CONUS that have been examined in previous studies. Al Bitar et al. (2012)
conducted an extensive evaluation of SMOS data using two networks we utilized: the Soil Climate Analysis
Network (SCAN; 20.32 cm) and SNOwpack TELemetry (SNOTEL; 20.32 cm). Additionally, we obtained soil
moisture observations from two other CONUS networks: the U.S. Department of Energy Atmospheric Radiation
Measurement (ARM; 25 cm) program (Jackson et al 1999) and the U.S. Climate Reference Network (USCRN; 20
cm; Bell et al., 2013). Complete ARM observations only existed from eras 1 to 4 and USCRN data was available for
only era 5. *In situ* values were aggregated to a daily time step (based on UTC time) that matched the surface
satellite-based soil moisture product described below. Figures 1 and 2 show the location of the stations selected
across the five eras.

The ARM network used the Campbell Scientific1 229-L heat dissipation matric potential sensor to estimate

soil moisture (Reece 1996). Calibration of this method was based on comparison of matric potential with soil water
release curves (Klute, 1986). Conversely, the SCAN, SNOTEL, and USCRN networks all used a Stevens Water
Hydra Probe (Schaefer et al., 2007; Bell et al., 2013). Seyfried et al. (2005) described the calibration approach and
how the dielectric measurements from the Hydra Probe sensor were converted into volumetric soil moisture
measurements.
**2.3 Surface Satellite-Based Soil Moisture**
This study was supported by four surface (5 cm) soil moisture products, three of which came from the CCI program.
We used the CCI Passive, CCI Active, and CCI Combined products. The harmonization process involved in the





creation of these products was described by Liu et al. (2012) and these datasets are available on-line
(http://www.esa-soilmoisture-cci.org/node/145). In addition, we also utilized stand-alone data from the AMSR-E
mission during eras 2-4. In this study we acquired the version produced by the Land Surface Parameter Model
(LPRM; Owe et al. 2008; ftp://hydrol.sci.gsfc.nasa.gov/data/s4pa/WAOB). All of these satellite soil moisture
products were produced at a daily time step with a 0.25º spatial resolution.
**2.4 Other Datasets**
Several other dataset were used in an ancillary role. Air temperature and precipitation data were obtained from
Parameter elevation Regression on Independent Slopes Model (PRISM; Daly et al. 1994) from grid cells (4 km
spatial resolution) co-located with examined *in situ* sites (PRISM Climate Group 2015). These data were used to
screen dates below freezing and with significant precipitation data, as suggested by (Dorigo et al., 2011), to enhance
quality control.

In addition, Normalized Difference Vegetation Index (NDVI) values (Tucker 1979) were used to help

constrain the only unknown in the exponential filter, the characteristic time length and was derived from Moderate
Resolution Imagining Spectroradiometer (MODIS) data. The version of MODIS (MOD13Q1) used near-infrared
reflectances that were atmospherically corrected to mask water, clouds, aerosols and cloud shadows. Datasets were
provided in a sinusoidal grid with a 250 m resolution and an average of nine pixels around each *in situ* station were
used to calculate a global average NVDI for each era.
**3 Methods**
**3.1 Initial Station Filtering**
To ensure selection of the highest quality *in situ* stations, we applied two criteria in our initial station selection. The
first criterion involved the amount of missing data within a candidate station. Sites that had an excessive number of
missing data, a total of over 20 days per year, were rejected. A second criterion related to a fundamental assumption
of the exponential filter method, which is that there is a hydrologic connection between the surface and root zone
horizons. One would expect that deeper within the profile there would be a greater lag in response. Therefore, a
linear correlation coefficient (r) between surface measurements (generally made at 5 cm) and lagged root zone data
from 20 to 25 cm depth was made. Root zone lag was calculated between 1 to 40 days and the day with the highest




correlation coefficient was selected. Stations whose maximum lagged correlation coefficient (r) fell below 0.5 were
rejected. Qiu et al. (2014) used a similar selection criterion in their study.
**3.2 Exponential Filter**
Wagner et al. (1999) originally developed the exponential filter and Albergel et al. (2008) refined this approach with
a more robust recursive version of this method. This version provided an estimate of a soil wetness index (SWI)
within the root zone. This index standardized RZSM based on the total range of values recorded by the *in situ*
dataset. The recursive formulation provided a predictor of RZSM at time ($t_n$), which in this study was given in days,
and was derived as:
$$SWI_{mn} = SWI_{mn(n-1)} + K_n \left[ ms(t_n) - SWI_{mn(n-1)} \right] \tag{1}$$
where $SWI_{mn(n-1)}$ represented the estimated RZSM at time $t_{n-1}$, $ms(t_n)$ was the surface soil moisture estimate based
on either CCI products or AMSR-E retrievals, and $K_n$ was the gain at time $t_n$ determined with:
$$K_n = \frac{K_{n-1}}{K_{n-1} + e^{\frac{t_n - t_{n-1}}{T}}} \tag{2}$$
where T represented the timescale of soil moisture variation in days. At the beginning of each era and after
excessively large gaps in $ms(t_n)$ data (> 12 days) the filter was initialized with $SWI_{m(1)} = ms(t_n)$ and $K_{n1}$ set to one.
Results from a data denial experiment described below provided support for the selection of 12 days as an
appropriate timescale to reset the filter. The prime advantage of the exponential filter was that the only unknown
was T.
**3.3 Objective Metrics**
Direct comparisons were made between CONUS *in situ* stations that represented a long-time series. While it is true
that soil moisture measurements exhibit a high degree of spatial variability over a wide range of spatial scales from
field plot to watershed (*e.g,* Western *et al*., 2004; Wilson *et al*., 2004; Brocca *et al*., 2007) temporal variation is
much more muted. Temporal stability is a concept fully rooted in soil science (Vachaud et al., 1985; Martinez-
Fernandez and Ceballos, 2003). Therefore, the approach of this study was to use standard objective metrics such as
correlation to describe the relationship between (coarse-scale) of root zone soil moisture estimates based on the





exponential filter and (point-scale) *in situ* measurements. Other temporal statistics included: bias, Nash-Sutcliffe
coefficients (NS), and root mean square error (RMSE, in volumetric soil moisture). Each of these metrics has their
own utility as discussed in the paper below.
**3.4 Calibration of $T_{opt}$**
Albergel et al. (2008) noted no significant correlation between soil properties and the optimal timescale of soil
moisture variation ($T_{Opt}$). Therefore, they constrained this parameter by optimizing T based on the NS metric, an
approach also applied by Ford et al. (2014). However, Albergel et al. (2008) also noted a weak relationship between
T with climate. Specifically, a linkage between increased temperatures and, hence, soil evaporation (not
transpiration). A lower $T_{Opt}$ was representative of a faster response of SWI present in areas with a higher
evaporational demand. This conjecture was consistent with a relationship developed by Qiu et al. (2014) using mean
NDVI values at *in situ* sites.
In this study we used two approaches to determine $T_{Opt}$. The first method optimized $T_{Opt}$ at a time in which
the RMSE is minimized. This was essentially the same approach as finding a maximum NS value. RMSE was
calculated between 1 to 68 days at a one-day increment. Sites that converged on the upper 68-day bound were
rejected. Qiu et al. (2014) used a similar upper bound as a means of selecting SCAN sites for their study.
The second approach used the NDVI formulation from Qiu et al. (2014) to calculate $T_{Opt}$. This relationship
is given as:
$T_{Opt} = [ -75.263 \text{ X NDVI} ] + 68.171$ (3)
**3.5 *In Situ* Station Filtering and Data Denial Experiment**
To ensure that the exponential filter was effective in producing a RZSM estimate, the ms ($t_n$) term was set based on
surface (5 cm) *in situ* data instead of satellite data. Normally grid based satellite surface moisture estimates are used
to drive the exponential filter. However, to establish a filter based on the quality of *in situ* data an initial estimate of
RZSM is determined based on surface *in situ* data at the 5 cm level. Initial RZSM estimates with a NS value less
than 0.50, which is a common threshold for defining a satisfactory match between *in situ* and simulated hydrologic
data (Moriasi *et al.,* 2007), were rejected. This filter removed many of the poor performing outliers (NS < -1.00)





from consideration. Table 1 describes the issues with the remaining poor performing outliers that lingered after this
*in situ* based filtering approach.

Use of surface (5 cm) *in situ* data also supported a data denial experiment that gauged how the filter's

performance was impacted by gaps in the ms ($t_n$) time series. This experiment focused on the SCAN network during
era 3 (2005-2008). Time series were altered to include only data at 2, 5, 8, and 11-day intervals. This experiment
was based on the 32 out of 42 sites that had *in situ* based NS in excess of 0.50; i.e. the sites that survived this
filtering process. Both surface (5 cm) *in situ* and satellite (AMSR-E) were used in this experiment.
**3.6 Spurious Data Filtering**
After calculation of rescaled SWI values for all four satellite products at each *in situ* station, a final series of filters
were applied to remove any spurious results following the qualify control guidelines articulated by Dorigo et al.
(2013). Surface temperature and precipitation data from co-located PRISM grid cells flagged problematic dates
within the time series of each dataset. Days in which the minimum air temperature was less than 0 ºC were removed
from the final rescaled SWI dataset. Satellite soil moisture retrieval were particularly fraught with difficulty under
freezing conditions (Dorigo et al., 2011). Likewise precipitation can be problematic and days with greater than 1
mm / day were excised following the guidance of (Dorigo et al., 2013).  Three additional flags related to the quality
of the *in situ* data were applied. Days with values in excess of the porosity reported by the ISMN were expunged
from the rescaled SWI dataset. Likewise, days that recorded the same value (plateaus) or zero were deemed spurious
and removed. Also, if the final filtered rescaled SWI dataset consisted of less than 100 days this dataset was rejected
following the guidance of Dorigo et al. (2013). Finally, SWI based estimates in which NS < -1.00 were rejected as
outliers. A detailed discuss of these outliers is given below.
**4 Results**
Figure 3 shows the results of the data denial experiment in which both *in situ* and satellite data (AMSR-E) was used
at the surface. Note a baseline performance for *in situ* dataset has an average NS values close to 0.7, which was
almost identical to results based on *in situ* surface soil moisture datasets in which every other day was withheld.
Even in datasets with every four out of five dates withheld there was only a slight drop in performance. This result
underscored the ability of the exponential filter to effectively cope with datasets that have significant gaps. Average



NS values fell to 0.5 only when over ninety percent of the surface soil moisture dataset was withheld and
measurements from only every eleventh day were used. Data denial experiment using AMSR-E data to drive the
filter yielded a similar drop-off in performance as the number of withheld days increased.
Figures 1 and 2 show lag correlation (r) between *in situ* surface (5 cm) and RZSM (20 to 30 cm) during the
five eras. ARM sites clustered in Oklahoma and Kansas had higher correlation coefficients during era 1 (Network
Average r = 0.864) and a drop in this metric during eras 2 to 4 (Network Average r = 0.793 to 0.796). SCAN sites
exhibited correlation coefficients that varied spatially. In general, better performances were noted from eastern
(Network Average r = 0.751 to 0.872) and central sites (Network Average r = 0.812 to 0.874). Western sites had
slightly lower r values (Network Average r = 0.699 to 0.770). Notable outliers were present for the stations in
Montana during eras 4 and 5 (Fig. 2) that could account partly for the poorer performance noted during these eras.
SNOTEL stations were concentrated in western CONUS and had consistently high correlation coefficients (Network
Average r = 0.828 to 0.865). Finally the USCRN sites examined during era 5 generally had better r values in eastern
and central CONUS (Network Average r = 0.846 to 0.882) as opposed to the west (Network Average r = 0.768).
The remainder of this section focuses on the results from the exponential filter driven by the four satellite
products. The $T_{Opt}$ and lagged r-values discussed are based on results that have a low absolute bias ($\pm$ 10%). As
might be expected, the $T_{Opt}$ values from the NDVI approach had a much more limited range of values compared
with $T_{Opt}$ values derived using the optimization approach (Tables 2 to 5). From the ARM network average $T_{Opt}$ based
on the NDVI approach ranged from 32 to 36 days whereas optimization produced much greater variation (4 to 32
days; Table 2). At SCAN the NDVI approach yielded a broader range of average era $T_{Opt}$ (28 to 46 days; Table 3).
But again optimization produced more variable $T_{Opt}$ values (9 to 39 days; Table 3). A similar pattern was noted at
SNOTEL sites. The NDVI approach yielded higher network average era $T_{Opt}$ values (42 to 45 days) versus the more
variable and lower results from the optimization method (17 to 36 days; Table 4). Finally, USCRN sites from era 5
exhibited a broad range of values for both approaches (NDVI = 30 to 55 days; Optimization = 9 to 28 days; Table

5).

Tables 2 to 5 show results from the direct correlation between *in situ* RZSM and SWI based estimates
generated from the four satellite products. Network average values are excluded in this discussion if there were less
than three measurements within an era for a network. Generally, but not always, the optimization approach yielded
higher lagged r-values than NDVI. Interestingly, in the ARM network in 5 out of 14 instances the NDVI approach




yielded network average r values that were greater than those obtained from the optimization method (Table 2).
ARM sites from the central Great Plains had network average r values based on optimization that ranged from 0.450
to 0.707 across eras 1 to 4; whereas the NDVI approach yielded a lower and broader variation in r values (0.323 to
0.704; Table 2).

For SCAN sites comparisons were made only for eras 2 to 5 (Table 3). Era 1 was excluded in this

comparison due to limited data availability during this period. Network average r-values based on optimization
(0.458 to 0.720; Table 3) generally outperform those based on the NDVI approach (0.428 to 0.615; Table 3).
Additionally, when examined from a geographic prospective, western CONUS sites had slightly higher r values
based on optimization (0.477 to 0.823) than those from either the east (0.332 to 0.777) or central regions (0.492 to

0.717).

SNOTEL stations from the intermountain west showed the greatest variability. Some sites recorded r-

values below 0, but there were also quite a few sites with high correlation coefficients (> 0.75). However, in general,
network average r-values were lower in SNOTEL (optimization = 0.370 to 0.572; NVDI = 0.228 to 0.590) than at
SCAN western sites (Table 4). Finally, the data from USCRN sites during era 5 had higher network average r-values
in central sites versus western CONUS (Table 5).

NS values across the five eras were depicted in Figs. 4-6. Stations with low absolute bias ($\pm$ 10%)

consistently outperformed stations with high bias within all networks and during all eras. This was true for both the
optimization and NDVI (data not shown) approaches to constraining T. Not surprisingly the optimization approach
generally outperformed the NDVI method. Also, the four satellite products had quite consistent results and did not
exhibit any clear temporal trends. All NS and RMSE network averages described below were based on the
optimization approach to constraining T and had a low absolute bias. Figure 4 showed NS results from the ARM and
USCRN networks. Network average NS values for ARM ranged from -0.1 to 0.3, similar to the results from the
USCRN network (0.2 to 0.3). Network average NS values from the SCAN and SNOTEL networks were shown in
Figs. 5 and 6, which were slightly higher (0.1 to 0.5).

Figures 7-9 depicted RMSE values again across the five eras. In many respects RMSE mirrors NS as a

performance metric. Like NS stations, RMSE values with a low absolute bias outperformed those with high bias.
However, the difference between low and high bias datasets was generally not as pronounced for the RMSE metric
as it was for NS. But like with NS, RMSE results showed no discernable temporal trends. RMSE values from the



ARM and USCRN networks were illustrated in Fig. 7. Network average RMSE values for ARM ranged from 0.02 to
0.04 and were significantly lower than values from the other networks examined in this study. USCRN network
average RMSE values ranged from 0.04 to 0.05 (Fig. 7). Figure 8 illustrated results from the SCAN network and
network average RMSE values were similar to USCRN sites (0.04 to 0.06). Finally, SNOTEL RMSE results (Fig. 9)
were higher than all other networks (0.05 to 0.07).
**5 Discussion and Conclusions**
A long-standing goal of the soil remote sensing community has been to develop techniques that can observe changes
in RZSM. Regrettably, the technology had not yet progressed to support a global RZSM product based only on
remote sensing retrievals. The use of land surface models such as the community NOAH model (Chen et al., 1996),
Global Land Data Assimilation System (GLDAS; Rodell et al., 2004), and European Centre for Medium-Range
Weather Forecasts (ECMWF) Re-analysis products (Uppala et al., 2005) have been used to fill this gap in recent
years. These platforms have become popular and provide an estimate of root zone soil moisture that has been
applied to field scale studies (Albergel et al. 2012; Blankenship et al. 2016; Kedzior et al. 2016). In addition, another
approach that has been suggested is based on thermal infrared based remote sensing *(e.g.* Hain et al., 2011).

Besides ease of use the exponential filter methodology is an attractive alternative because it leverages

existing remotely sensed soil moisture platforms. As such, this approach is not hindered by the incipit assumptions
built in to every modeling platform and relies purely on observational data. Given the potential utility of the
exponential filter approach, a detailed analysis of the potential errors associated with the method is in order. There
are four main sources of error. Two of these errors are associated with the SWI estimate and included:  (1) the
unsuitable of the exponential filter at a given site and (2) retrievals errors in the surface soil moisture dataset. The
other two errors are not related to the actual SWI estimate but instead are errors in the independent datasets that
were applied to verify the SWI estimate at the scale of the 0.25º satellite grid. These errors included: (3) issues with
*in situ* datasets (Dorigo et al. 2011, 2013) and (4) non-representativeness of a point site when compared with the
large (0.25º) footprint of a surface soil moisture grid used to drive the filter (Crow et al. 2012). A significant quality
control measure involved driving the filter with surface *in situ* instead of satellite soil moisture data. Stations that
scored a NS < 0.5 based on this approach were rejected as not suitable. At these sites perhaps the fundamental
assumption of the exponential filter method that there was hydrologic equilibrium between and the surface and root


zone was violated. Therefore, the gross errors recorded at some sites cannot be ascribed to issues with the
exponential filter and the data denial experiment demonstrated the robustness of this method at least in certain
instances (Fig. 3).

Analysis of poor performing outliers (NS < 1.00) provided additional insights into how the exponential

filter can fail at some sites (Table 1). Within the ARM network all outliers could be attributed to *in situ* data issues
such as spikes, breaks, anomalous high values that exceed soil porosity, anomalous low values at zero, and extended
plateaus (Dorigo et al. 2013). An example of such a clearly flawed *in situ* dataset is shown in Fig. 10 a. Within the
SNOTEL network there was more of a mix in error type (Table 1). Besides *in situ* data issues, another significant
source of error was the limited number of days in some of the final SWI datasets. Following the guidance of Dorigo
et al. (2010) SWI datasets with less than 100 days were rejected. However, based on observations in this study,
significant issues of representativeness were noted when there were less than 400 days (Fig. 10 b). The high altitude
of many SNOTEL sites resulted in a longer freezing season during which a greater number of days were filtered out.
There were some sites with *in situ* data issues in the SCAN network (Table 1). However, many of the outliers also
were caused by either SWI values that lacked the dynamic range of the *in situ* dataset (Fig. 10 c) or SWI values that
had significant timing offsets compared with *in situ* RZSM observations (Fig. 10 d). These issues were the result of
either site non-representativeness or errors in surface soil moisture retrievals. Finally, USCRN sites exhibited a
similar mix of errors as noted in the SCAN network (Table 1).

A consistent result noted in this study was the impact of bias on other performance metrics. Consistently

better results for all metrics were noted (Tables 2-5; Figs. 4-9) when there was a low absolute bias (within 10%)
versus SWI datasets that had a high absolute bias (>10%). Additionally, this observation was observed for SWI
values produced with both approaches to constrain T (minimization of RMSE and NDVI approach). The impact of
bias on standard objective metrics was a focus of temporal stability analysis (Vachaud et al., 1985; Martinez-
Fernandez and Ceballos, 2003). Sites with little variation in bias yielded more robust comparisons with remote
sensing data (Starks et al., 2006); a result that was confirmed in this study across four distinct *in situ* soil moisture
networks and satellite products.

Interestingly, the results observed in this study were more impacted by the *in situ* network than the surface

satellite product used to drive the exponential filter. In terms of the NS metric, SCAN, SNOTEL, and USCRN
outperformed ARM (Figs. 4-6). The NS metric seemed to have a greater utility in indentifying outliers than the



RMSE metric. This was because it ranged from 1.00 to potentially -∞, unlike RMSE, which ranged in this study
from only 0 to 0.14.

Conversely, when considering the RMSE metric, ARM sites yielded superior scores compared with SCAN,

SNOTEL, and USCRN (Figs. 7-9). Within the ARM network average RMSE was less than 0.04, which is the
baseline value for accuracy designed for many satellite soil moisture missions (*e.g.* Kerr et al. 2001; Entekhabi et al.,
2010). SCAN and USCRN were slightly above this guideline and were similar to RMSE values noted in previous *in*
*situ*/satellite soil moisture comparisons (*e.g.* Brocca *et al.*, 2010; Jackson *et al.*, 2010, 2012; Al Bitar *et al.*, 2012).
According to the RMSE metric SNOTEL sites performed the worst and was significantly above the 0.04
performance target.

Perhaps the most interesting result from this study was that the performance metrics in each *in situ* network

did not vary over time. Given that almost two decades of data examined, this finding is particularly noteworthy.
Therefore SWI estimates of RZSM produced by the exponential filter using CCI datasets can be leveraged for long-
term, perhaps even multi-decadal, climate studies (Manfreda et al., 2011). Another fruitful line of future research
could compare exponential filter estimates of RZSM with those generated by land surface models. With the
proliferation of space-based remote sensing platforms and the continued development of *in situ* monitoring networks
the duration of RZSM time series will only grow. As such, the approaches outlined in this work can provide the
cornerstone to support future assessments of long-term trends in RZSM, which is an essential climate variable.
*Acknowledgements.* We acknowledge the support of the NASA Climate Indicator and Data Products for Future
National Climate Assessments program through award # NNX16AH30G. The assistance of Robert Parinussa
(University of New South Wales), Arturo Diaz (Texas A&M International University), and Luis Carrasco Garza
(Texas A&M International University) is greatly appreciated.

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

501                                  **TABLES**

Table 1. Number of poor performing (NS < 1.00) outliers for all four satellite products.
RMSE Optimization

|  | **ARM** | **SCAN** | **SNOTEL** | **USCRN** |
|---|---|---|---|---|
| *In situ* Data | 17 | 3 | 15 | 1 |
| Insufficient SWI | 0 | 1 | 14 | 0 |
| Lack of Range | 0 | 11 | 0 | 3 |
| Timing Issues | 0 | 0 | 9 | 0 |


NDVI Approach

|  | **ARM** | **SCAN** | **SNOTEL** | **USCRN** |
|---|---|---|---|---|
| *In situ* Data | 22 | 16 | 32 | 5 |
| Insufficient SWI | 0 | 3 | 44 | 0 |
| Lack of Range | 0 | 17 | 15 | 8 |
| Timing Issues | 0 | 6 | 5 | 3 |







Table 2. Average lagged correlation factor (r) and $T_{Opt}$ between SWI based and *in situ* soil
moisture at the 25 cm depth for the ARM network. Standard derivation is indicated in
parentheses.

Optimization Approach – Low Bias
**AMSR-E**    **CCI-Combined**    **CCI-Passive**    **CCI-Active**

| Era | n | r value | $T_{opt}$ | n | r value | $T_{opt}$ | n | r value | $T_{opt}$ | n | r value | $T_{opt}$ |
|---|---|---|---|---|---|---|---|---|---|---|---|---|
| 1 | --- | -------- | ---- | 14 | 0.471 (0.249) | 30 (19) | 4 | 0.614 (0.131) | 25 (29) | 9 | 0.450 (0.193) | 26 (13) |
| 2 | 9 | 0.587 (0.080) | 4 (1) | 10 | 0.491 (0.136) | 9 (4) | 10 | 0.554 (0.103) | 7 (6) | 11 | 0.493 (0.153) | 17 (7) |
| 3 | 12 | 0.589 (0.148) | 7 (3) | 12 | 0.520 (0.156) | 12 (10) | 12 | 0.615 (0.165) | 8 (4) | 12 | 0.460 (0.165) | 13 (10) |
| 4 | 4 | 0.666 (0.053) | 32 (10) | 3 | 0.707 (0.081) | 10 (4) | 2 | 0.649 (0.011) | 12 (1) | 1 | 0.823 | 5 |


NDVI Approach– Low Bias
**AMSR-E**    **CCI-Combined**    **CCI-Passive**    **CCI-Active**

| Era | n | r value | $T_{opt}$ | n | r value | $T_{opt}$ | n | r value | $T_{opt}$ | n | r value | $T_{opt}$ |
|---|---|---|---|---|---|---|---|---|---|---|---|---|
| 1 | | -------- | ---- | 17 | 0.439 (0.241) | 36 (3) | 9 | 0.480 (0.171) | 36 (2) | 12 | 0.414 (0.172) | 36 (4) |
| 2 | 7 | 0.622 (0.156) | 35 (3) | 11 | 0.567 (0.172) | 34 (4) | 9 | 0.642 (0.132) | 34 (4) | 13 | 0.484 (0.154) | 32 (3) |
| 3 | 13 | 0.559 (0.204) | 34 (2) | 12 | 0.437 (0.179) | 35 (3) | 10 | 0.645 (0.137) | 34 (3) | 12 | 0.341 (0.197) | 34 (3) |
| 4 | 5 | 0.666 (0.053) | 32 (6) | 3 | 0.704 (0.004) | 34 (2) | 3 | 0.665 (0.542) | 34 (2) | 7 | 0.323 (0.184) | 32 (3) |







Table 3. Average lagged correlation factor (r) and $T_{Opt}$ between SWI based on optimization and
*in situ* soil moisture at the 20.32 cm depth for the SCAN network (Figures 1 and 2). Standard
derivation is indicated in parentheses.

Optimization Approach – Low Bias
**AMSR-E**          **CCI-Combined**        **CCI-Passive**        **CCI-Active**

| Era | n | r value | $T_{opt}$ | n | r value | $T_{opt}$ | n | r value | $T_{opt}$ | n | r value | $T_{opt}$ |
|-----|---|---------|-----------|---|---------|-----------|---|---------|-----------|---|---------|-----------|
| 1 | --- | -------- | ---- | 1 | 0.817 | 19 | 1 | 0.691 | 1 | 3 | 0.458 (0.323) | 22 (10) |
| 2 | 4 | 0.691 (0.157) | 39 (19) | 7 | 0.598 (0.157) | 27 (16) | 2 | 0.661 (0.007) | 16 (9) | 7 | 0.519 (0.147) | 15 (6) |
| 3 | 17 | 0.596 (0.129) | 10 (7) | 19 | 0.556 (0.164) | 14 (13) | 16 | 0.556 (0.184) | 9 (5) | 17 | 0.521 (0.140) | 17 (17) |
| 4 | 14 | 0.697 (0.096) | 15 (14) | 16 | 0.698 (0.155) | 19 (15) | 10 | 0.720 (0.176) | 15 (12) | 16 | 0.642 (0.226) | 17 (16) |
| 5 | --- | -------- | ---- | 17 | 0.572 (0.183) | 16 (15) | 11 | 0.472 (0.192) | 21 (14) | 15 | 0.589 (0.195) | 14 (14) |


NDVI Approach– Low Bias
**AMSR-E**          **CCI-Combined**        **CCI-Passive**        **CCI-Active**

| Era | n | r value | $T_{opt}$ | n | r value | $T_{opt}$ | n | r value | $T_{opt}$ | n | r value | $T_{opt}$ |
|-----|---|---------|-----------|---|---------|-----------|---|---------|-----------|---|---------|-----------|
| 1 | --- | -------- | ---- | 2 | 0.678 (0.199) | 32 (6) | 2 | 0.747 (0.096) | 49 | 4 | 0.463 (0.282) | 40 (10) |
| 2 | 6 | 0.554 (0.198) | 34 (16) | 7 | 0.541 (0.179) | 30 (12) | 1 | 0.330 | 20 | 10 | 0.505 (0.171) | 28 (7) |
| 3 | 14 | 0.596 (0.111) | 31 (10) | 15 | 0.480 (0.193) | 34 (11) | 15 | 0.613 (0.095) | 36 (11) | 15 | 0.471 (0.187) | 31 (10) |
| 4 | 16 | 0.573 (0.242) | 37 (15) | 20 | 0.585 (0.223) | 39 (15) | 14 | 0.615 (0.238) | 39 (15) | 20 | 0.608 (0.226) | 40 (15) |
| 5 | --- | -------- | ---- | 19 | 0.518 (0.220) | 39 (13) | 15 | 0.428 (0.238) | 46 (11) | 26 | 0.469 (0.237) | 41 (13) |






Table 4. Average lagged correlation factor (r) and $T_{Opt}$ between SWI based on optimization and
*in situ* soil moisture at the 20.32 cm depth for the SNOTEL network. Standard derivation is
indicated in parentheses.

Optimization Approach – Low Bias

| | AMSR-E | | | CCI-Combined | | | CCI-Passive | | | CCI-Active | | |
|---|---|---|---|---|---|---|---|---|---|---|---|---|
| Era | n | r value | $T_{opt}$ | n | r value | $T_{opt}$ | n | r value | $T_{opt}$ | n | r value | $T_{opt}$ |
| 2 | 5 | 0.572 (0.311) | 17 (15) | 2 | 0.600 (0.034) | 10 (1) | 2 | 0.750 (0.054) | 14 (7) | 3 | 0.509 (0.156) | 36 (13) |
| 3 | 39 | 0.463 (0.264) | 20 (15) | 17 | 0.513 (0.290) | 27 (18) | 30 | 0.461 (0.293) | 25 (20) | 30 | 0.370 (0.317) | 29 (11) |
| 4 | 63 | 0.508 (0.299) | 18 (14) | 32 | 0.491 (0.353) | 20 (16) | 55 | 0.522 (0.302) | 18 (11) | 32 | 0.522 (0.379) | 22 (18) |
| 5 | --- | -------- | ---- | 5 | 0.527 (0.189) | 25 (13) | 12 | 0.412 (0.252) | 26 (17) | 8 | 0.534 (0.319) | 27 (21) |


NDVI Approach– Low Bias

| | AMSR-E | | | CCI-Combined | | | CCI-Passive | | | CCI-Active | | |
|---|---|---|---|---|---|---|---|---|---|---|---|---|
| Era | n | r value | $T_{opt}$ | n | r value | $T_{opt}$ | n | r value | $T_{opt}$ | n | r value | $T_{opt}$ |
| 2 | 2 | 0.678 (0.197) | 44 (13) | 1 | 0.438 | 49 | 4 | 0.584 (0.102) | 45 (8) | 4 | 0.444 (0.362) | 44 (7) |
| 3 | 44 | 0.367 (0.374) | 44 (6) | 28 | 0.313 (0.395) | 44 (7) | 43 | 0.334 (0.386) | 44 (6) | 45 | 0.327 (0.337) | 44 (5) |
| 4 | 71 | 0.425 (0.367) | 43 (6) | 33 | 0.385 (0.491) | 43 (7) | 61 | 0.451 (0.341) | 44 (7) | 41 | 0.228 (0.529) | 44 (6) |
| 5 | --- | -------- | ---- | 11 | 0.425 (0.216) | 44 (7) | 9 | 0.357 (0.318) | 43 (5) | 10 | 0.590 (0.268) | 42 (6) |







Table 5. Average lagged correlation factor (r) and $T_{Opt}$ between SWI based on optimization and

*in situ* soil moisture at the 20 cm depth for the USCRN network during era 5. Standard derivation

is indicated in parentheses. Sites are divided by region (east, central, west) as indicated on Figure

2.

Optimization Approach – Low Bias

| | **CCI-Combined** | | | **CCI-Passive** | | | **CCI-Active** | | |
|---|---|---|---|---|---|---|---|---|---|
| **Region** | **n** | **r value** | **$T_{opt}$** | **n** | **r value** | **$T_{opt}$** | **n** | **r value** | **$T_{opt}$** |
| East | 1 | 0.105 | 4 | -- | -------- | ---- | 1 | 0.486 | 15 |
| Central | 13 | 0.594 (0.185) | 9 (8) | 6 | 0.707 (0.086) | 17 (19) | 11 | 0.607 (0.126) | 6 (3) |
| West | 1 | 0.857 | 11 | 4 | 0.406 (0.125) | 28 (21) | 3 | 0.540 (0.389) | 9 (1) |

NDVI Approach– Low Bias

| | **CCI-Combined** | | | **CCI-Passive** | | | **CCI-Active** | | |
|---|---|---|---|---|---|---|---|---|---|
| **Region** | **n** | **r value** | **$T_{opt}$** | **n** | **r value** | **$T_{opt}$** | **n** | **r value** | **$T_{opt}$** |
| East | 2 | 0.388 (0.122) | 1 | 1 | 0.071 | 25 | 2 | 0.410 (0.133) | 21 |
| Central | 12 | 0.521 (0.231) | 30 (10) | 7 | 0.605 (0.194) | 35 (9) | 7 | 0.534 (0.176) | 25 (7) |
| West | 3 | 0.209 (0.068) | 36 (20) | 4 | 0.342 (0.128) | 45 (20) | 3 | 0.087 (0.122) | 55 (5) |





**Figure Captions**

Figure 1. Locality map of examined *in situ* stations (ARM - X; SCAN - ∗; SNOTEL - +) with (**a**) era 1, (**b**) era 2, and (**c**) era 3.

Figure 2. Locality map of examined *in situ* stations (ARM - X; SCAN - ∗; SNOTEL - +) with (**a**) era 4 and (**b**) era 5. During era 5 (X) represents USCRN instead of ARM stations.

Figure 3. Box plot of data denial experiment from the SCAN network during era 3 (2005-2008). Results for day 1 represent baseline data for the exponential filter driven by surface soil moisture data (*in situ* data – stars; low absolute bias RMSE optimized AMSR-E – circles). Other time series were altered to include only data at 2, 5, 8, and 11-day intervals.

Figure 4. Box plots that depict the NS metric for the ARM (eras 1 to 4) and USCRN (era 5) networks. Results for high absolute bias RMSE optimized datasets are squares, low absolute bias RMSE optimized datasets are circles, and low absolute bias NVDI datasets are triangles.

Figure 5. Box plots depicting NS metric for the SCAN network. Symbols are as in Figure 4.

Figure 6. Box plots depicting NS metric for the SNOTEL network. Symbols are as in Figure 4.

Figure 7. Box plots depicting RMSE metric for the ARM (eras 1 to 4) and USCRN (era 5) networks. Symbols are as in Figure 4.

Figure 8. Box plots depicting RMSE metric for the SCAN network. Symbols are as in Figure 4.

Figure 9. Box plots depicting RMSE metric for the SNOTEL network. Symbols are as in Figure 4.

Figure 10. Selected time series associated with poorly performing (NS < 1.00) outliers with *in situ* data as solid gray and SWI estimates in dashed black. (**a**) Shows an example of problematic *in situ* data. (**b**) Is an example where there was insufficient SWI data. (**c**) Illustrates an SWI dataset that lacked the dynamic range present in the *in situ* data. (**d**) Depicts a discrepancy in timing between SWI and *in situ* datasets.




























**FIGURE 1**

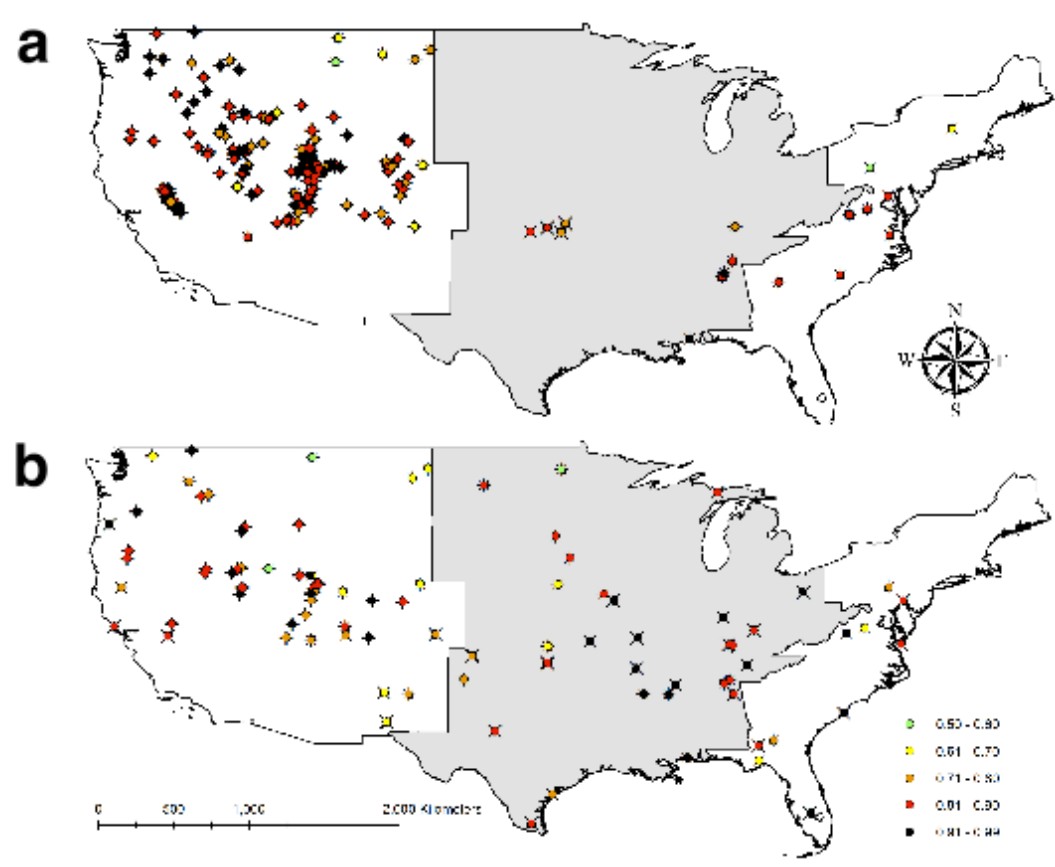



**FIGURE 2**


















# Figure 3




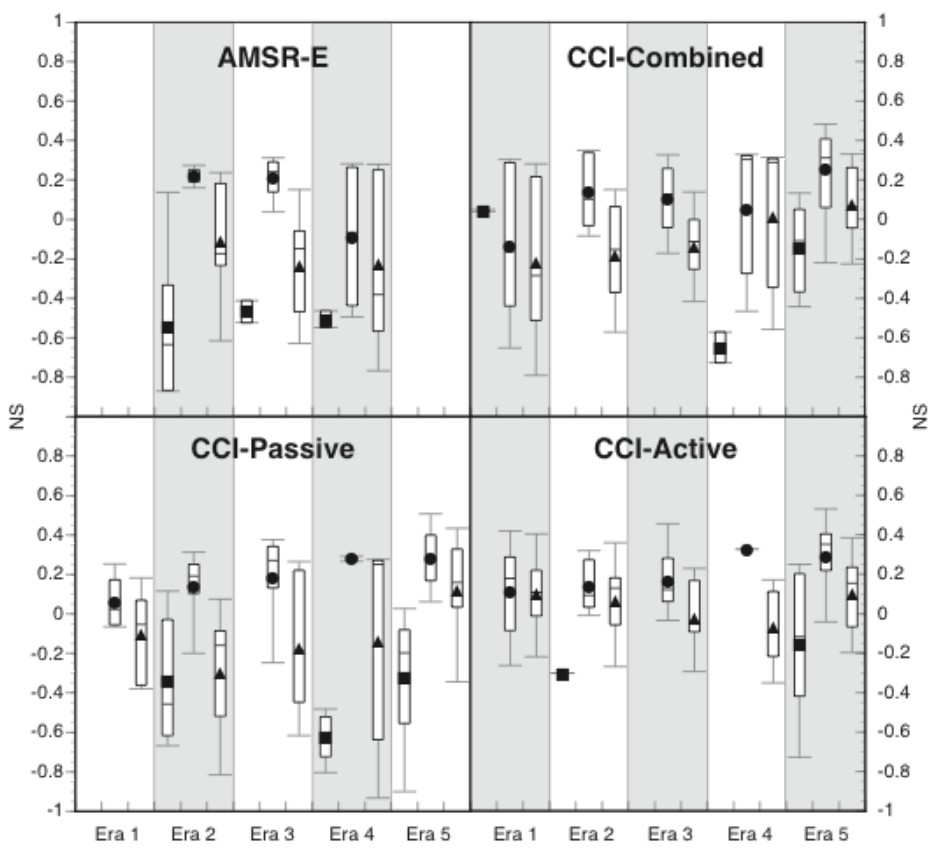

**Figure 4**






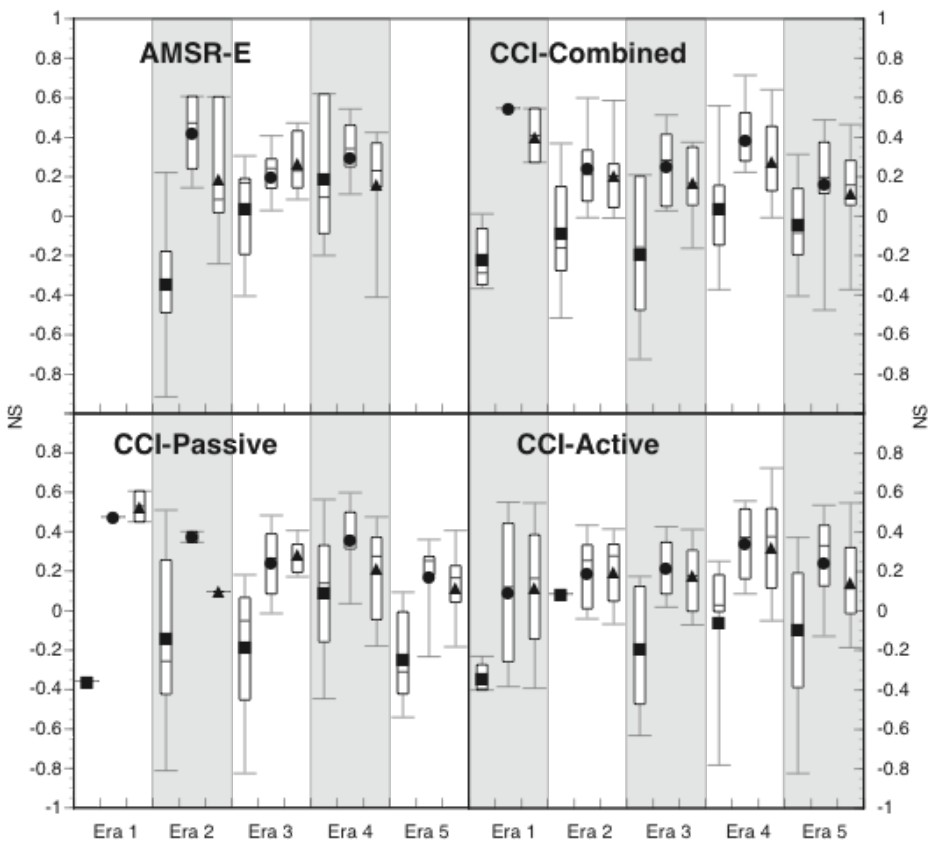

## Figure 5






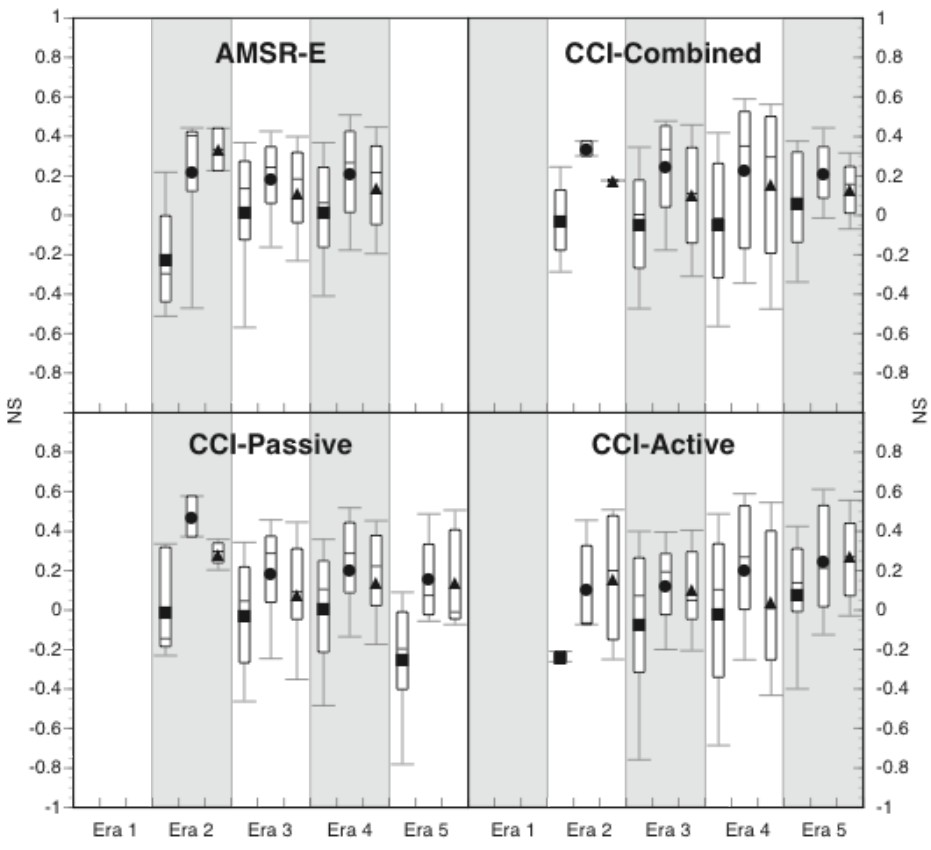

Figure 6






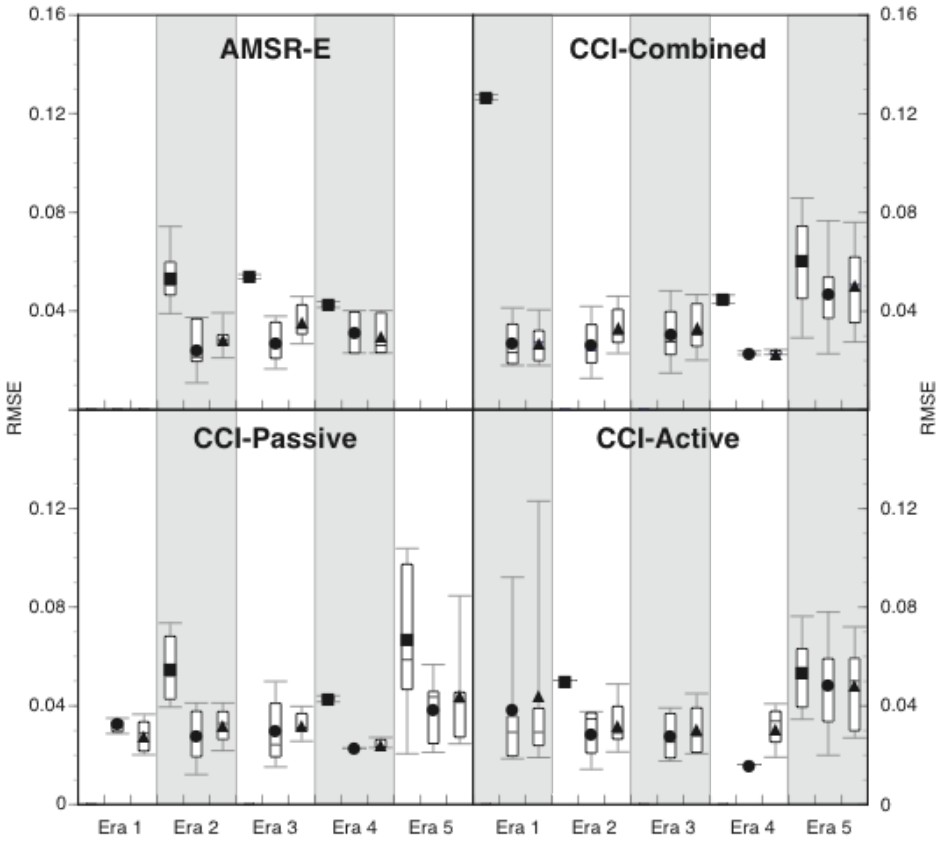

**Figure 7**





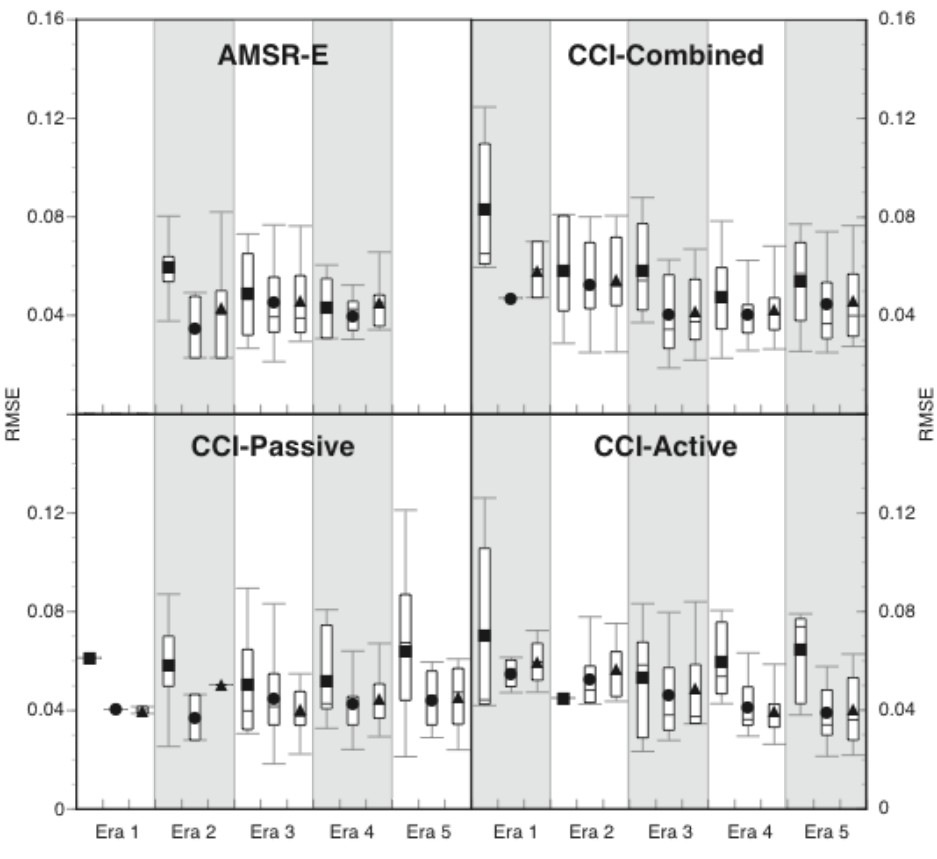

**Figure 8**






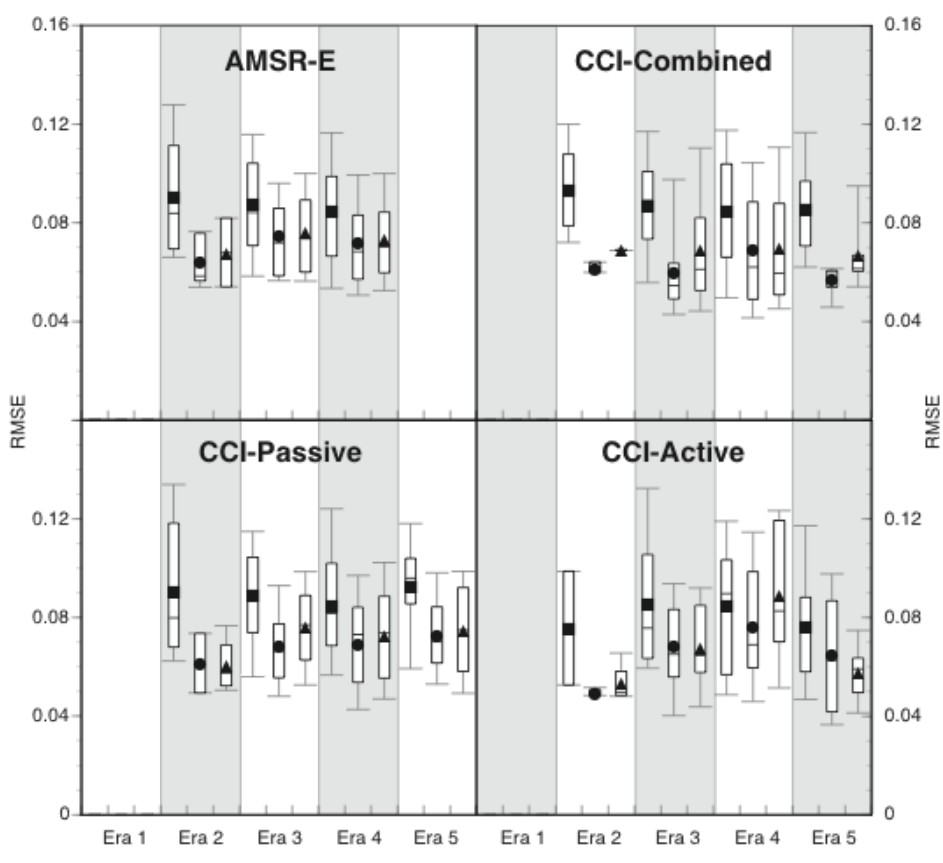

**Figure 9**






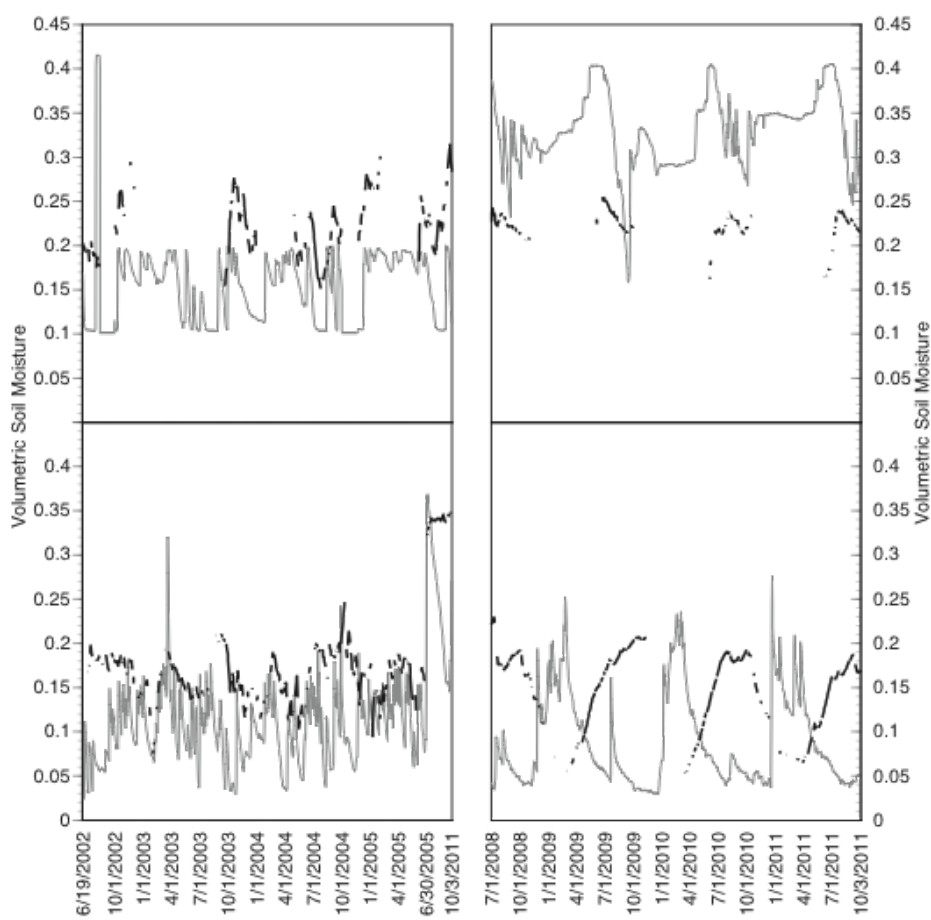

**Figure 10**
