# Peer review of "Multi-decadal analysis of root-zone soil moisture applying the exponential filter across CONUS"

_Hydrology and Earth System Sciences, 2017_

## Referee Comment (RC1) · Anonymous Referee #1 · 9 Jun 2017

Review of the paper:

**"Multi-decadal analysis of root-zone soil moisture applying the exponential filter across CONUS"**

**by: Kenneth J. Tobin, Roberto Torres, Wade T. Crow, and Marvin E. Bennett**

**GENERAL COMMENTS**

The paper applies the exponential filter to produce an estimate of root-zone soil moisture (RZSM) from four types of microwave-based, surface satellite soil moisture products, namely: the NASA LPRM AMSR-E product and 3 additional products obtained from the European Space Agency Climate Change Initiative (CCI), i.e., the CCI-Active, CCI-Passive and the CCI-Combined soil moisture products.

The estimates are compared against in situ observations derived from 4 different networks (AMR, SCAN, SNOTEL, USCRN) in Contiguous United STATES (CONUS) during 4 eras selected according AMSR-E data availability.

The recursive formulation of the exponential filter (Albergel et al. 2008) is used for the evaluation of the Soil Water Index. The calibration of the only parameter T is carried out in two ways: 1) by the minimization of the RMSE against in situ root zone soil moisture and 2) by using NDVI derived from MODIS as in Qiu et al. (2014).

The topic is of interest for the HESS readership and worth to be published. The paper is also well organized and contains interesting analyses and results. I particularly appreciate the use of the NDVI index for the filter calibration which I think should be more highlighted and more discussed in the paper. Despite this, I found some moderate issues that the authors should address prior the paper can be considered adequate for publication. My main comments are listed below:

- 1. The distinction among the different eras is done by considering AMSRE soil moisture product as a reference (pre-AMSRE, post-AMSRE etc...). Given the paper results, it is also clear that the gaps contained in the surface soil moisture data have an effect on the performance of the exponential filter (although this effect is limited for gaps < 2days). Based on that, I wonder if the rationale in the choice of the different eras should not take into consideration also the advent of the ASCAT sensor. Indeed, according to the paper of Dorigo et al. 2015 (Figure 5) it can be observed a significant increase in the data density after 2007, i.e., when the Advanced Scatterometer ASCAT has started to be operational. From the figure, we can also observe a significant reduction of the data density during 2001-2003, which might have an effect on the performance of the exponential filter (especially after the quality check, which might significantly reduce the data density).
- 2. It is not completely clear from the paper how the problem of gaps in the satellite soil moisture data is addressed within the application of the exponential filter. I suppose the authors use the exact time difference between two valid satellite observations, i.e., tn-tn-1. When this time difference is large (even weeks in northern CONUS), as it might happen during the winter season, is the filer re-started? Please add a brief discussion on that.
- 3. The quality of the figures is not appropriate and some captions are not self-explanatory. This makes difficult the interpretation of the results. I will list below specific comments on them.

- 4. The version of the specific CCI product is not mentioned. There are sensible changes between the different versions of the products with respect to the merging procedure and the sensors used (e.g. SMOS) that justify the inclusion of the product version in the manuscript.
- 5. At a point of the manuscript (line 256 pag 10) it reads: "The  $T_{Opt}$  and lagged r-values discussed are based on results that have a low absolute bias (±10%)". Nothing is said before about any distinction between stations with low and high absolute bias and why this distinction is included in the results. Nor the authors provide the number and which of the stations show low and high absolute bias. I found this confusing. This issue of the bias is also confounding with the notation used at page 9 line 224 where SWI is defined as "rescaled SWI". Do the authors use any rescaling technique? If so, how it is related with the bias? Which rescaling technique has been used?

Based on the comment above I recommend the publication after MODERATE REVISIONS.

I will list below my specific comments in order of appearance in the manuscript also indicating their relevance.

| PAGE | LINES       | RELEVANCE | COMMENT                                                                                                                                                                                                                                                                                                                                                                                                                                                                                                                                           |
|------|-------------|-----------|---------------------------------------------------------------------------------------------------------------------------------------------------------------------------------------------------------------------------------------------------------------------------------------------------------------------------------------------------------------------------------------------------------------------------------------------------------------------------------------------------------------------------------------------------|
| 4    | 103-
109 | MINOR     | Why not add a table of the selected eras. This would simply the reading.                                                                                                                                                                                                                                                                                                                                                                                                                                                                          |
| 5    | 136         | MODERATE  | Indicate the CCI version here. This is important.                                                                                                                                                                                                                                                                                                                                                                                                                                                                                                 |
| 9    | 227         | MOD/MAJOR | "Days in which the minimum air temperature was less
than 0 °C were removed from the SWI dataset." This
is an important point. I think it is not completely
correct to remove these values at this point (i.e., after
the application of the filter). Indeed, given the
recursive nature of the filter any surface observation
characterized by a temperature lower than 0°C has a
detrimental effect on future observations. Hence, this
masking has to be carried out on the surface
observations and not on the SWI. |
| 10   | 245         | MINOR     | Here the authors uses "lag correlation", in section 3.3 simply "correlation" while in the figures "lagged correlation factor". Later lagged r-values. Please use a consistent notation.                                                                                                                                                                                                                                                                                                                                                           |
| 10   | 256         | MOD/MAJOR | "low absolute bias" (see point 5 of the main comments)                                                                                                                                                                                                                                                                                                                                                                                                                                                                                            |
| 11   | 285         | MODERATE  | See previous point                                                                                                                                                                                                                                                                                                                                                                                                                                                                                                                                |
| 12   | 310         | MINOR     |  <li>You may also cite this work (Massari et al. 2014) in which the authors used ERA-Interim root zone soil moisture within a simple hydrological model to infer the catchment wetness conditions before flood events.</li> <li>Massari, C., Brocca, L., Barbetta, S., Papathanasiou, C., Mimikou, M., & Moramarco, T. (2014). Using globally available soil moisture indicators for flood modelling in Mediterranean</li>                                                                                                           |

|                   |     |          | catchments. Hydrology and Earth System
Sciences , 18 (2), 839–853. Retrieved from
http://www.scopus.com/inward/record.url?eid=
2-s2.0-
84896859292&partnerID=40&md5=02d91d85
22bb7c834f88e95a0266c9a3 |
|-------------------|-----|----------|-------------------------------------------------------------------------------------------------------------------------------------------------------------------------------------------------------------------------------------------|
| 13                | 328 | MINOR    | "NS<1" should it be NS<-1                                                                                                                                                                                                                 |
| Table2
caption |     | MINOR    | Define n in the table                                                                                                                                                                                                                     |
| Figure
1       |     | MODERATE | Improve the legend quality and provide explanation about grey areas.                                                                                                                                                                      |
| Figure
10      |     | MODERATE | Add letters to identify subfigures and add a legend.                                                                                                                                                                                      |

---

## Referee Comment (RC2) · Anonymous Referee #2 · 9 Jun 2017

The paper shows an important systematic study of the application of the exponential filter to estimate SWI and RZSD from various SM satellite products. The results are of great interest for research and applications that use these products and prefer a simple and reliable method such as SWI for estimating RZSD. However, the conclusions are limited to commenting on the different performance of the SWI method comparing RZSD estimates and in situ measurements, using a series of literature metrics (N-S, bias, RMSE). I think the paper conclusions would surely be more effective if it could include further insights in the discussion e.g. linking the failure of the exponential filter in some areas and under some conditions, tying the behavior to hydrological/morphological factors, e.g. The nature/type of the soil, the landscape, LU/LC, etc.

---

## Author Comment (AC1) · 20 Jun 2017

Anonymous Referee #1 Main Comments: 1. The distinction among the different eras is done by considering AMSRE soil moisture product as a reference (pre-AMSRE, post-AMSRE etc. . .). Given the paper results, it is also clear that the gaps contained in the surface soil moisture data have an effect on the performance of the exponential filter (although this effect is limited for gaps < 2days). Based on that, I wonder if the rationale in the choice of the different eras should not take into consideration also the advent of the ASCAT sensor. Indeed, according to the paper of Dorigo et al. 2015 (Figure 5) it can be observed a significant increase in the data density after 2007, i.e., when the Advanced Scatterometer ASCAT has started to be operational. From the figure, we can also observe a significant reduction of the data density during 2001-2003, which

might have an effect on the performance of the exponential filter (especially after the quality check, which might significantly reduce the data density).Âă

Response: One of the surprising results from this study is that the objective metrics generally did not show much variation over time (lines 363 to 364). Therefore, I do not believe that changing the era definitions would have a great impact on the results of this study. The availability of the ASCAT sensor would only impact results using CCI-Active and CCI-Combined data in some instances. Redefinition of eras would have no effect on RZSM estimates based on AMSR-E or CCI-Passive.

2. It is not completely clear from the paper how the problem of gaps in the satellite soil moisture data is addressed within the application of the exponential filter. I suppose the authors use the exact time difference between two valid satellite observations, i.e., tn-tn-1. When this time difference is large (even weeks in northern CONUS), as it might happen during the winter season, is the filer re-started? Please add a brief discussion on that.Âă

Response: I believe I mentioned that when gaps extend beyond 12 days the filter is reset (lines 181 to 182). This time length is consistent with the major drop-off in objective functions evident during our data denial experiment. Yes, the filter is reset (K=1) when the gaps are greater than 12 days.

3. The quality of the figures is not appropriate and some captions are not self-explanatory. This makes difficult the interpretation of the results. I will list below specific comments on them.

Response: We will improve the figure quality and legends particularly for Figures 1 and 10 and Table 2 as suggested.

4. The version of the specific CCI product is not mentioned. There are sensible changes between the different versions of the products with respect to the merging procedure and the sensors used (e.g. SMOS) that justify the inclusion of the product

version in the manuscript.Âă

Response: We will provide information about the specific version of CCI used in this study in the revised paper.

5. At a point of the manuscript (line 256 pag 10) it reads: "The TOpt and lagged r-values discussed are based on results that have a low absolute bias ($\pm$ 10%)". Nothing is said before about any distinction between stations with low and high absolute bias and why this distinction is included in the results. Nor the authors provide the number and which of the stations show low and high absolute bias. I found this confusing. This issue of the bias is also confounding with the notation used at page 9 line 224 where SWI is defined as "rescaled SWI". Do the authors use any rescaling technique? If so, how it is related with the bias? Which rescaling technique has been used?Âă

Response: More information about the rescaling procedure will be provided in the revised paper which is quite a simple method commonly used in the soil moisture community. As for the mention of low versus high bias this actually turned out to be an important distinction in data quality where low bias results nearly always outperformed sites exhibiting higher bias. More explanation of the impact of bias on the presented results will be offered in the revised paper.

Specific Comments: A. Why not add a table of the selected eras. This would simply the reading.Âă

Response: This will be added to the revised paper.

B. Indicate the CCI version here. This is important.Âă

Response: This will be added to the revised paper.

C. "Days in which the minimum air temperature was less than 0 $^\circ$C were removed from the SWI dataset." This is an important point. I think it is not completely correct to remove these values at this point (i.e., after the application of the filter). Indeed, given the recursive nature of the filter any surface observation characterized by a temperature

lower than 0$^\circ$C has a detrimental effect on future observations. Hence, this masking has to be carried out on the surface observations and not on the SWI.Âă

Response:

D. Here the authors uses "lag correlation", in section 3.3 simply "correlation" while in the figures "lagged correlation factor". Later lagged r-values. Please use a consistent notation.Âă

Response: We will correct the notation in the revised paper.

E & F. "low absolute bias" (see point 5 of the main comments)Âă.... See previous pointÂă

Response: See response to main comment #5 above.

G. You may also cite this work (Massari et al. 2014) in which the authors used ERA-Interim root zone soil moisture within a simple hydrological model to infer the catchment wetness conditions before flood events.Âă

Response: We will add this reference to the revised paper.

H. "NS<1" should it be NS<-1Âă

Response: This will be fixed in the revised paper.

I to K Define n in the tableÂă …. Improve the legend quality and provide explanation about grey areas.Âă… Add letters to identify subfigures and add a legend.

Response: See response to main comment #3 above.

Anonymous Referee #2 The paper shows an important systematic study of the application of the exponential filter to estimate SWI and RZSD from various SM satellite products. The results are of great interest for research and applications that use these products and prefer a simple and reliable method such as SWI for estimating RZSD. However, the conclusions are limited to commenting on the different performance of

the SWI method comparing RZSD estimates and in situ measurements, using a series of literature metrics (N-S, bias, RMSE). I think the paper conclusions would surely be more effective if it could include further insights in the discussion e.g. linking the failure of the exponential filter in some areas and under some conditions, tying the behavior to hydrological/ morphological factors, e.g. The nature/type of the soil, the landscape, LU/LC, etc.

Response: In the revised paper will more closely explore in more detail why the exponential filter failed in certain conditions by examining the slope, land cover, soil types, etc. in the satellite soil moisture grid that is collocated with the in situ station.